# Empirical Bayes functional models for hydrogen deuterium exchange mass spectrometry

Oliver M. Crook [1✉], Chun-wa Chung[2] & Charlotte M. Deane [1]

Hydrogen deuterium exchange mass spectrometry (HDX-MS) is a technique to explore differential protein structure by examining the rate of deuterium incorporation for specific peptides. This rate will be altered upon structural perturbation and detecting significant changes to this rate requires a statistical test. To determine rates of incorporation, HDX-MS measurements are frequently made over a time course. However, current statistical testing procedures ignore the correlations in the temporal dimension of the data. Using tools from functional data analysis, we develop a testing procedure that explicitly incorporates a model of hydrogen deuterium exchange. To further improve statistical power, we develop an empirical Bayes version of our method, allowing us to borrow information across peptides and stabilise variance estimates for low sample sizes. Our approach has increased power, reduces false positives and improves interpretation over linear model-based approaches. Due to the improved flexibility of our method, we can apply it to a multi-antibody epitope-mapping experiment where current approaches are inapplicable due insufficient flexibility. Hence, our approach allows HDX-MS to be applied in more experimental scenarios and reduces the burden on experimentalists to produce excessive replicates. Our approach is implemented in the R-package "hdxstats": https://github.com/ococrook/hdxstats.

[1] Department of Statistics, University of Oxford, Oxford OX1 3LB, UK. [2] Structural and Biophysical Sciences, GlaxoSmithKline R&D, Stevenage SG1 2NY, UK.
✉email: oliver.crook@stats.ox.ac.uk

Probing a protein structure is essential to fully understand its function[1]. Protein structures can be perturbed when binding to another protein, small molecule or due to alterations in their context (such as pH)[2–4]. Hydrogen deuterium exchange (HDX) mass spectrometry is one such technique to examine differential context-specific protein structure[5,6]. The guiding hypothesis is that, when a protein is incubated with heavy water, amide hydrogens exchange with deuterium in accordance with Linderstom-Lang theory[7]. Though a number of factors, such as solvent occlusion, topological flexibility, amino acid content and secondary structure, also affect the process[8–11]. By monitoring the kinetics of HDX using bottom-up mass spectrometry, it is possible to discern subtle alterations to a protein's structure[3]. To mediate possibly complex kinetics the process is usually examined over a time-course. However, this temporal component is rarely used in statistical testing of alterations to the kinetics e.g.,[12–19].

A number of approaches have been proposed to detect differences in peptide HDX between samples. These include manual examination of the data, linear models, linear mixed-models and a student's t test amongst others[12–19]. The most statistically advanced approach to analyse differential HDX is termed MEMHDX[16], which introduced random effects to previous linear modelling approaches[13]. Models incorporating both fixed effects and random effects are called mixed effects, these model excel at modelling nested variance according to the replicate structure of the data[20]. However, they can be difficult to interpret for unseasoned users. Furthermore, MEMHDX suggest to examine p-values for magnitude of deuterium change and change in dynamics. This subtly conflates significance with effect size and we suggest to examine a number of effects alongside the p-value to accurately interpret the nature of the kinetic changes.

All currently proposed methods avoid explicitly modelling the temporal component of HDX data, which reduces statistical power. In experiments with low samples sizes, such as HDX-MS where only a handful of measurements are made per peptide, maximising power is crucial to infer condition-specific differences. Hence, methodology that models serial correlations will improve power in HDX-MS experiments with low samples sizes. Statistics based on sampled curves, so-called *functional data analysis*, concerns the analysis of such functions[21]. Here, we turn the functional model of HDX into a statistical test, namely a functional analysis of variance (ANOVA). This approach is simpler to interpret and more powerful than previously proposed methods. The advantages arise because of a reduction in the number of parameters and tests performed, whilst also allowing the model to capture serial correlations[21]. Furthermore, we can exploit repeated measurement across different peptides to estimate a pooled variance. The estimated sample variance for each peptide can then be shrunk towards this pooled variance, resulting in more stable estimates of variance when the sample size is small. This idea is called empirical Bayes and has been highly influential in the analysis of microarrays[22], RNA-seq[23] and proteomics experiments[24]. We establish this method for HDX data as applied to functional models, see[25] for applications to linear HDX models.

Our article is structured as follows. First, we compare a number of methodologies through simulated examples to demonstrate the effects of different methodological choices, replicate structure and number of time points. We then apply our approach to a number of real-world experiments and find that linear mixed models are unable to control false positives, whilst the t test is unable to declare any results significant. We then proceed to an epitope mapping experiment for which none of the current methods are applicable. We demonstrate that our approach is able to make quantitative statements in experimental scenarios, where other available methods are not. In particular, our method uncovers significantly

altered kinetics in an epitope mapping experiment applied to an HOIP-RBR, for which previous results have only been qualitative[26]. Our results support an observation that some single domain antibodies hold HOIP-RBR in a more open conformation[26].

Our approach is made freely available to the community through an R-package "hdxstats": https://github.com/ococrook/hdxstats, which builds on the "QFeatures" package for general quantitative mass-spectrometry[27].

## Results

**Simulations**. To obtain a more reliable statistical method for hydrogen-deuterium exchange mass spectrometry (HDX-MS), we developed an empirical Bayes functional model based on a proposed Weibull model for the kinetics (see methods). Our approach can be summarised as fitting a functional model that is blinded to the condition. The model is then refitted separately for each condition. By examining the residual error of each of these two scenarios, we can compute an F-statistic, which is then *moderated* by pooling information across peptides. The null hypothesis is that the condition-independent model is sufficient to explain the data. Figure 1a shows the fitted model kinetics where there is no difference between the two conditions, whilst panel b shows a situation where the kinetics significantly differ between the two conditions. From these fitted models we can compute an F statistic, high values of the F statistics indicate sufficient evidence to reject the null hypothesis. Such high values are obtained by examining the appropriate F distribution (see methods and Fig. 1c). Such an approach avoids testing each time point separately, explicitly incorporates serial correlations and improves power by stabilising variance estimates.

We compare our method to other major approaches (MEMHDX[16] and t tests). t tests are an exact test for the equality of means of two populations. In HDX-MS, this corresponds to the application of the t test pointwise at each measured time point. If the *false discovery rate* (FDR) for a peptide is lower than some nominal value, say 0.05, we have sufficient evidence to say the HDX kinetics were perturbed at that time point. MEMHDX is a linear mixed modelling approach. These approaches are usually applied when multiple measurements are made on the same quantity of interest, which induces a structured variance. In the case of MEMHDX, replicates are encoded as a random effect. If the coefficients of the linear model related to the condition (see methods) are declared significant, then this would be taken as evidence of perturbed HDX kinetics. We note that t tests are a special case of linear mixed models with one level and random intercepts. Since the t test and linear mixed models produce many p-values per peptide, they are combined to a single p-value using the harmonic mean p-value[28]. All methods are corrected for multiple testing using the Benjamini–Hochberg procedure[29].

In our simulation study (see methods), we are assessing whether methods can detect known perturbations to the HDX kinetics. If a method declares an FDR of less than 0.05 for a given peptide which has condition-dependent kinetics, then this is called a true positive. Whilst, if the FDR is less than 0.05 but the peptide does not have condition-dependent kinetics then that is defined as a false positive. Hence, methods are assessed using the F score (not to be confused with the F statistic), which weighs up precision and recall. In all cases, our approach outperforms the other methods except in the last simulation where only 1% of peptides were simulated to have significant differences, where it performed equally well to the t test. This performance level indicates that our approach can reliably detect differences even with only a couple of replicates and can handle HDX measurements that are missing at random. The t test performs poorly because it is difficult to accurately estimate the population

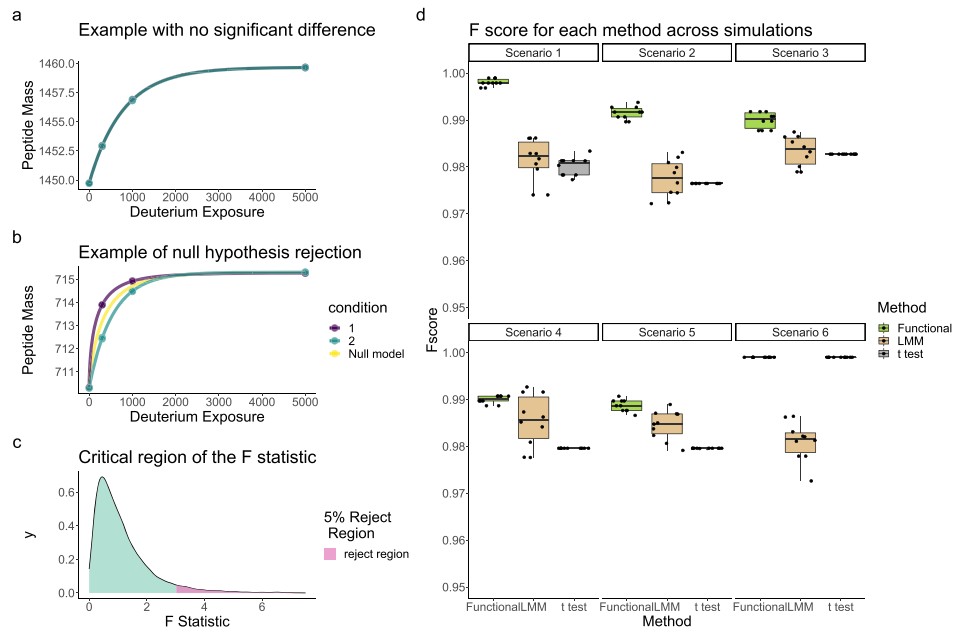

**Fig. 1 Functional model and performance in simulations. a–c** Cartoon of the proposed approach to statistical analysis of HDX-MS data. a) Functional models fit where there is no difference between the conditions. **b** Functional models fitted to HDX data where there are significant differences. **c** The F distribution with 5% critical region identified in magenta. **d** Simulation study for HDX data showing improved performance of our proposed functional method (functional) for linear mixed models (LMM) and the *t* test (*t* test). These simulations encompass scenarios with differing numbers of time points and replicates. We refer the reader to the methods for details.

standard deviation and thus it is underpowered. MEMHDX performs better than the *t* test because it models additional correlations via the random effect but because these correlations are not explicitly parametrised they are less powerful than our functional model. We explore simulations with higher levels of noise in the supplementary material (see Supplementary Fig. 18) and obtain the same conclusions (see Supplementary Note 6). In the next section, we demonstrate that MEMHDX also inflates false positives.

### Applications

*Structural variant experiment.* We compared each of the approaches in practice. Outside of simulations, true positives are not well defined; however, the methods can be tested on their ability to control false positives. We examined a previously published structural variant experiment, where HDX data on maltose-binding protein (MBP) was generated in seven replicates across four HDX labelling times[17]. Additional experiments were carried out in triplicate for the W169G (tryptophan residue 169 to glycine) structural variant. Here MBP-W169G was spiked into the wild-type MBP sample in 5, 10, 15, 20, 25% proportions, and a further experiment included a 100% mutant sample. All data were analysed on a Agilent 6530 Q-TOF mass spectrometer and raw spectra processed in HDExaminer.

The seven MBP samples without any structural variant can be used as a null experiment by partitioning the replicates *falsely* into two conditions. That is three of the samples are labelled condition A and four samples are labelled condition B, arbitrarily. We randomly permute the samples labelled A and B, six times. We then performed statistical significance testing between the conditions using the three methods previously considered. Since experiments are in fact all replicates, if the FDR is correctly controlled, there should be no peptides declared significant.

From Fig. 2 we see that our proposed method and the *t* test perform well at avoiding false positives, each generating only one false positive across the six permutation experiments. However,

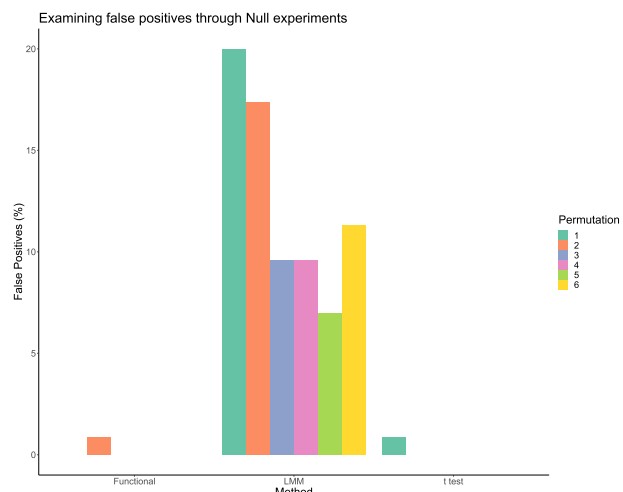

**Fig. 2 Performance of statistical methods in null experiments.** Our functional approach (functional), linear mixed models (LMM) and the *t* test (*t* test) are compared in null experiments.

the linear mixed modelling approach generates excessive false positives, with between 6% and 22% false positives per experiment. Here, the percentage is of the total peptides measured in the experiment. Hence, we can conclude that the linear mixed modelling approach is too liberal to be reliable in practice.

Our proposed functional statistical approach is built using a parametric functional model allowing us to interpret the statistical significance of our results, beyond simply pairwise differences. We compared wild-type MBP with the 100% structural variant sample, using all methods. Given there are seven replicates of the wild-type protein, statistical power (and henceforth, simply, power) is not an issue for any of the approaches. However, for most experiments, seven replicates are

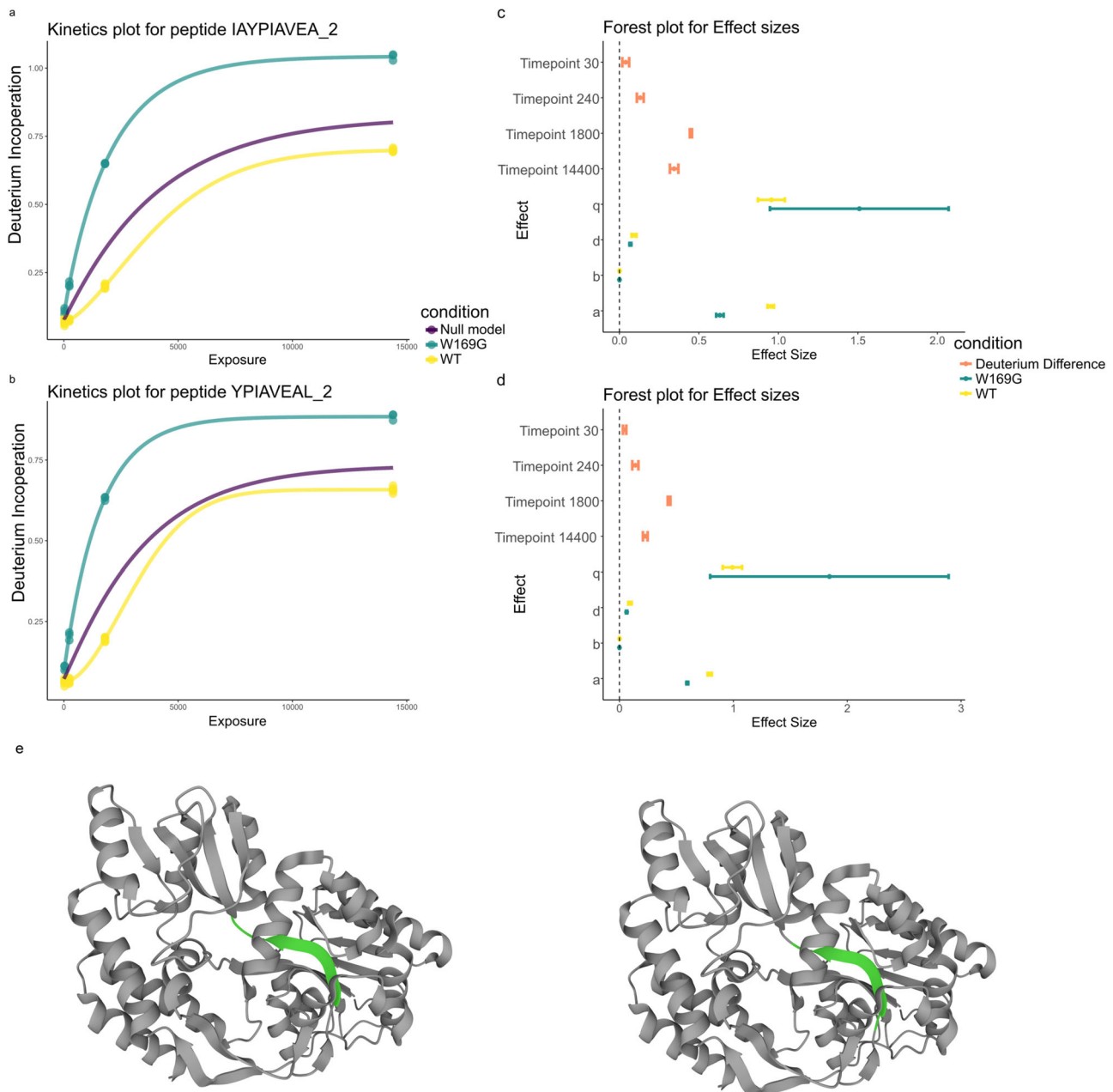

**Fig. 3 Functional models are interpretable. a** Deuterium kinetics for peptide IAYPIAVEA [119-138] in charge state two. Proportion of deuterium incorporated is plotted against solvent exposure time in seconds. Conditions are coloured with the null model in purple. The number after the underscore indicates the charge state. (**b**) same as for (**a**) but for peptide YPIAVEAL [131–139] in charge state one. **c** Forest plot for effect sizes corresponding to peptide in IAYPIAVEA [129–138]. **d** Forest plot for effect sizes, the strength of relationship between variables (see methods), corresponding to peptide in YPIAVEAL [131–139]. **c**, **d** The pointer refers to the mean estimate with tails corresponding to the 95% confidence interval. **e** MBP (PDB: 1OMP) with peptides highlighted in green YPIAVEAL [131–139] (left) IAYPIAVEA [129–138] (right).

likely to be onerous. Each parameter of our functional model can be interpreted and the magnitude of the parameters can be considered as effects sizes. We are particular interested in the parameters $b$ and $p$ of our functional model because they control the time-dependent kinetics (see methods for a full description). In particular, values of $p > 1$ suggest more rapid than exponential exchange of deuterium suggesting that a region has become more exposed to the solvent. Meanwhile, $b$ controls the rate of plateau, such that larger values of $b$ indicate the deuterium uptake plateaus more quickly. The parameter $a$ models the plateau itself, whilst $d$ models the intercept. A forest plot can be used to simultaneously visualise several possible effects that we might be

interested in to improve the interpretation of the results. Figure 3 shows two peptides with overlapping residues and their fitted models. In both cases the kinetics are significantly different between the WT and structural variant (FDR < $10^{-8}$) using our empirical Bayes $F$ test. Panels c and d of this figure show that the pairwise differences at each time point are different from 0. The parameters $d$ and $b$ only display small changes, whilst $a$, which models the curves' plateau, is significantly different. This suggests some residues for these peptides have become accessible due to the mutation compared with the WT protein. We can also see that $p \approx 1.5$ for peptide IAYPIAVEA [129–138] and $p \approx 1.8$ for YPIAVEAL [131–139] in the W169G variant, whilst for the WT

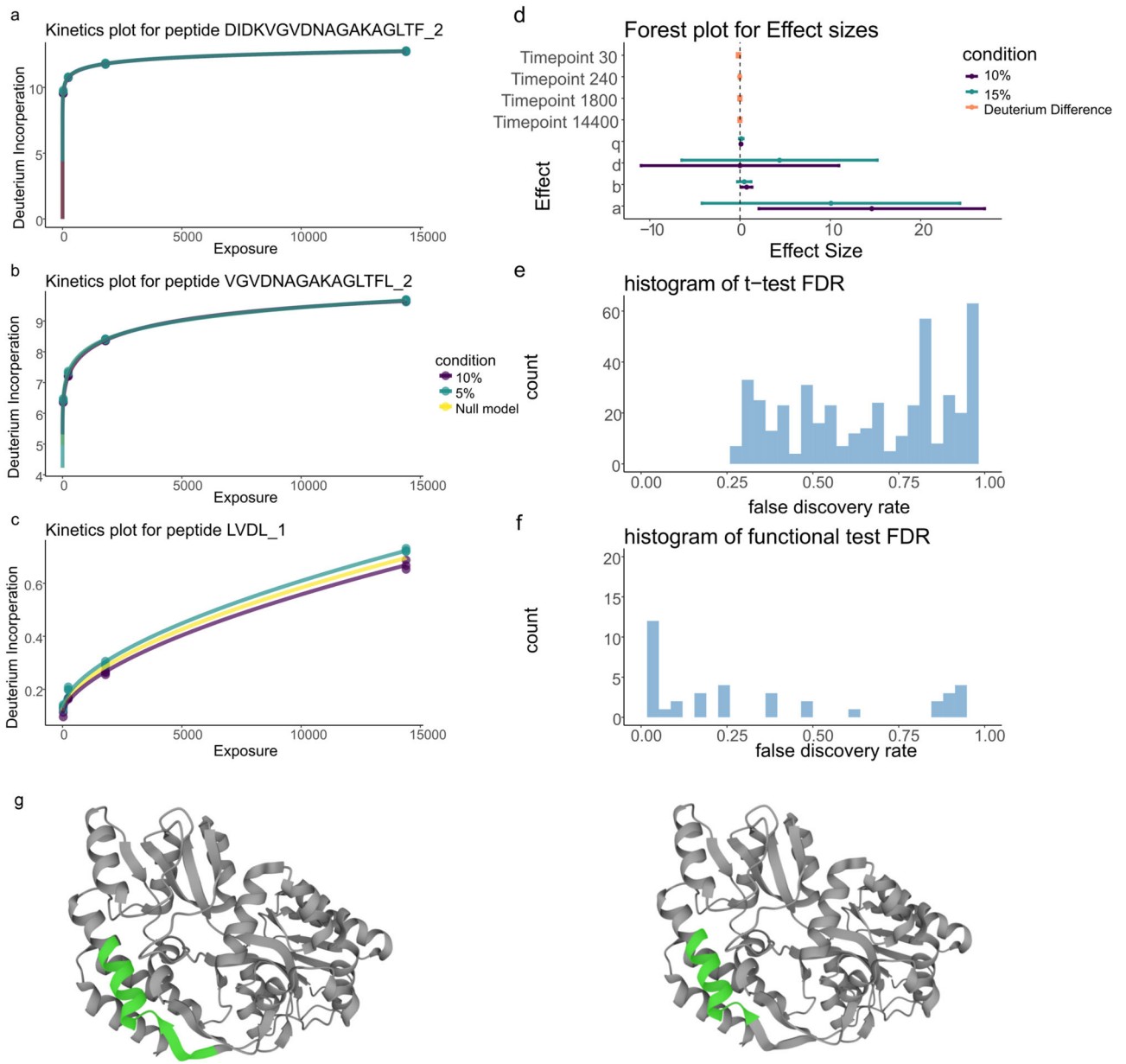

**Fig. 4 Functional models can identify subtle, consistent differences. a** Deuterium kinetics for peptide DIKDVGVDNAGAKAKAGLTF [203-220] in charge state 3. Deuterium incorporation is plotted against solvent exposure time in seconds. Conditions are coloured with the null model in yellow. The number after the underscore indicates the charge state. (**b**) same as for (**a**) but for peptide VGVDNAGAKAGLTFL [207–220] in charge state 2. (**c**) as for (**a**) but for peptide LVDL [221–224] in charge state 1 (**d**) Forest plot for effect sizes corresponding to peptide in (**a**). **e**, **f** Histogram of adusted *p*-values for the *t* test and functional method, correspondingly. **g** MBP (PDB: 1OMP) with DIKDVGVDNAGAKAKAGLTF (left) and VGVDNAGAKAGLTFL (right) highlighted in green.

$p \approx 1$ in both cases. This suggests more rapid than exponential exchange in deuterium for the mutant and further evidence that this region has become more solvent accessible and/or has fewer hydrogen bonds. The concerted behaviour between the two peptides adds further support to this hypothesis. These peptides are localised to a buried beta-strand suggesting the structural variant has a large impact on the protein's structure (Fig. 3e).

We then assessed our method on its ability to detect subtle differences in HDX experiments. We took the 10% and 15% structural variant samples from the structural variant experiments on MBP. We applied our functional model to the data and found that 12 peptides had an adjusted *p*-value smaller than 0.05. Three examples are plotted in Fig. 4a–c with the remainder plotted in the supplementary material. Figure 4d shows the forest plot

corresponding to the peptide in Fig. 4a. Here, the difference is visually subtle but our functional method identifies a difference between the two samples. Indeed, at all time points, the deuterium incorporation for the 10% sample is lower than that of the 15% structural variant sample. However, for three out of the four samples, the confidence intervals in this difference overlapped with 0 (see Table 1). The power of functional methods is that they can identify the consistency in an effect across time points, allowing us to identify significant changes that are consistent but not necessarily significant at any time point individually. This finding is reinforced with an overlapping peptide demonstrating concordant behaviour (Fig. 4b, g). The *t* test fails to find any significant differences at level 0.05 and the histogram of adjusted *p*-values is shown in Fig. 4e. The lack of

**Table 1 Difference between deuterum incoperation for peptide DIKDVGVDNAGAKAGLTF in the 10% and 15% structural variant experiments.**

|   | Estimate | confL | confU | Time point | condition |
|---|---|---|---|---|---|
| 1 | −0.17 | −0.31 | −0.03 | 30 | Change in deuterium uptake |
| 2 | −0.03 | −0.09 | 0.02 | 240 | Change in deuterium uptake |
| 3 | −0.02 | −0.14 | 0.11 | 1800 | Change in deuterium uptake |
| 4 | −0.01 | −0.13 | 0.11 | 14400 | Change in deuterium uptake |

Central estimates (mean) are reported alongside the 95% confidence intervals. The lower (upper) tail of the confidence interval is given by confL (confU).

uniformity and peaking of values toward 0 suggests low power, whilst the expected trend is seen for the functional method in Fig. 4f. Even deuterium differences that are apparent are not detected by the $t$ test (see 4c). Our functional models ability to detect differences whilst controlling false positives in both simulations and experiments suggests it is a more appropriate method for HDX-MS.

*Epitope mapping of HOIP-RBR using HDX-MS.* In this section, we explore an application of HDX-MS to epitope mapping of HOIP-RBR. HOIP is an E3 ubiquitin-protein ligase which conjugates linear polyubiquitin chains and plays a role in immune signalling[30–32]. Usually, binding epitopes are identified by "protection"; that is, surface amides that incorporate deuterium more slowly as they are shielded from the solvent[26] performed HDX-MS experiments for HOIP-RBR upon single domain antibody (dAb) complexation and in APO state. Mass-spectrometry was performed using a Waters Synapt G2-Si instrument and raw data was processed using DynamX. HDX-MS measurements were taken at 0, 30 and 300 s post-exposure to heavy water, for thirteen dAbs at different molar concentrations. Only a single replicate measurement was taken in each state so that more measurements of different dAbs could be made. However, this renders the $t$ test inapplicable because we cannot compute a within-group variance and linear mixed-models (MEMHDX) are inapplicable because there is not a nested replicate structure. However, it is still possible to apply our proposed functional method.

To avoid over-fitting, we fix $b = 0.5, q = 1, d = 0$ in our Weibull model, reducing the complexity of our model to a single degree of freedom in the null case. This greatly reduces the flexibility of our model, but in return we can apply rigorous statistical testing. These parameter choices roughly correspond to an assumption that 80% of deuterium will be incorporated within 30 seconds and the kinetics will plateau by 300 seconds.

For brevity, we focus on dAb25 from the study of ref. [26], because they observed non-standard HDX behaviour for this complex (the remainder are shown in the supplementary material). We applied our functional method as detailed in the methods, with the alterations described in the previous paragraph. We identified eight peptides for which the deuterium kinetics were altered (adjusted $p$-value < 0.05). six of these peptides displayed non-classical behaviour with increased deuterium incorporation on dAb binding (see Fig. 5). Three of these peptides overlap with each other and are contained within the helix-turn-helix (linker) region of HOIP (top row Fig. 5), whilst the other three also overlap and are contained in the RING2 region of HOIP (middle row Fig. 5)). The remaining two peptides also overlap with each other and are contained within the IBR (in-between ring) region of HOIP (final row Fig. 5). This suggests that the epitope for dAb25 is contained with the IBR region and this binding holds HOIP in a more open conformation allowing increased solvent exposure; hence, more deuterium exchange is possible.

To provide the spatial context for these changes in deuterium kinetics, we plotted a Manhattan plot; that is, peptides plotted against $-\log_{10}(p-\text{value})$ (see Fig. 6). This helps us to simultaneously visualise protein domain regions and (de)protected regions, as described in the previous paragraph. Whilst examining this plot, we also notice tendencies for $p$-values to cluster, suggesting correlations in the spatial axis of HDX data. This is expected, since these peptides either physically overlap or come from the same protein domain. Furthermore, we notice some of these clustered $p$-values fall just below the significance threshold suggesting power could be boosted by modelling correlations in this dimension as part of future work. Results for the remaining peptides and a residue-level analysis can be found in Supplementary Figs. 1–14.

## Discussion

We have presented an empirical Bayes functional data analysis approach for HDX-MS data. Our model explicitly incorporates the temporal component of these data, which boosts power and interpretation. Furthermore, we developed an empirical Bayes testing approach to stabilise variance estimates across the peptides in the experiment. The resulting methodology is more powerful than previous linear model-based approaches, as demonstrated by our simulation study. These earlier approaches lack power because they do look explicitly incorporate the temporal component nor borrow information across peptides.

We made an empirical comparison of the approaches in an application to structural variant data. This analysis concluded that the mixed-modelling approach was unable to control the false discovery rate, whilst our approach and the $t$ test were able to control it. However, application to a case with subtle differences demonstrated that the $t$ test was unable to declare any peptides significant. Hence, our approach controls false positives whilst providing peptides which can be followed up.

Having demonstrated the empirical statistical properties of our method, we applied our approach to a case study of multi-antibody epitope mapping of HOIP-RBR. Current approaches are not able to assess the significance in these experiments because of their stringent assumptions. However, our empirical Bayes functional method is applicable and was able to find significant differential HDX kinetics. Thus, we are able to identify the binding epitopes and allosteric effects of the single domain antibodies on HOIP-RBR.

Our approach has a number of limitations. Firstly, we do not model correlations in the spatial domain of HDX data - that is between overlapping peptides. This manifests as clustering of $p$-values for overlapped peptides. Several strategies exist to reconcile the spatial dimension of HDX data, including combining $p$-values using multi-level testing or spatial random effects, which we will consider if future work. Secondly, we do not model correlations in our multi-antibody study across the different antibody experiments. Joint modelling across related experiments is likely to boost power and interpretation further. Finally, our analysis

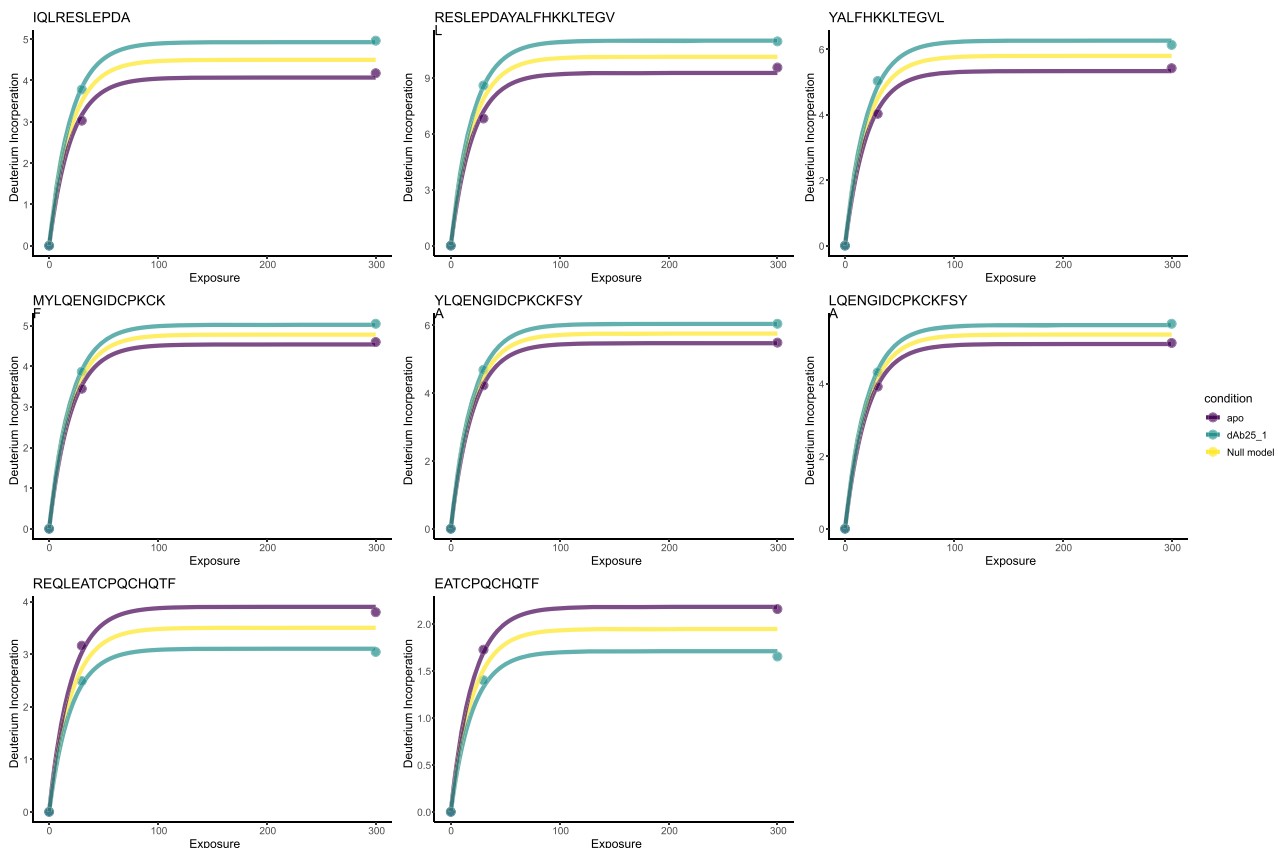

**Fig. 5 Functional model identifies protected and unprotected region of HOIP-RBR in complexation with dAb 25.** The top row contains kinetic plots for peptides in the helix-turn-helix region of HOIP; the middle row contains kinetic plots for peptides in the RING2 region of HOIP; the final row contains proteins in the IBR region of HOIP. Deuterium incorporation is plotted in units of Daltons and Exposure to heavy water in seconds. Yellow lines indicate the null model, whilst green and purple indicate the alternative model.

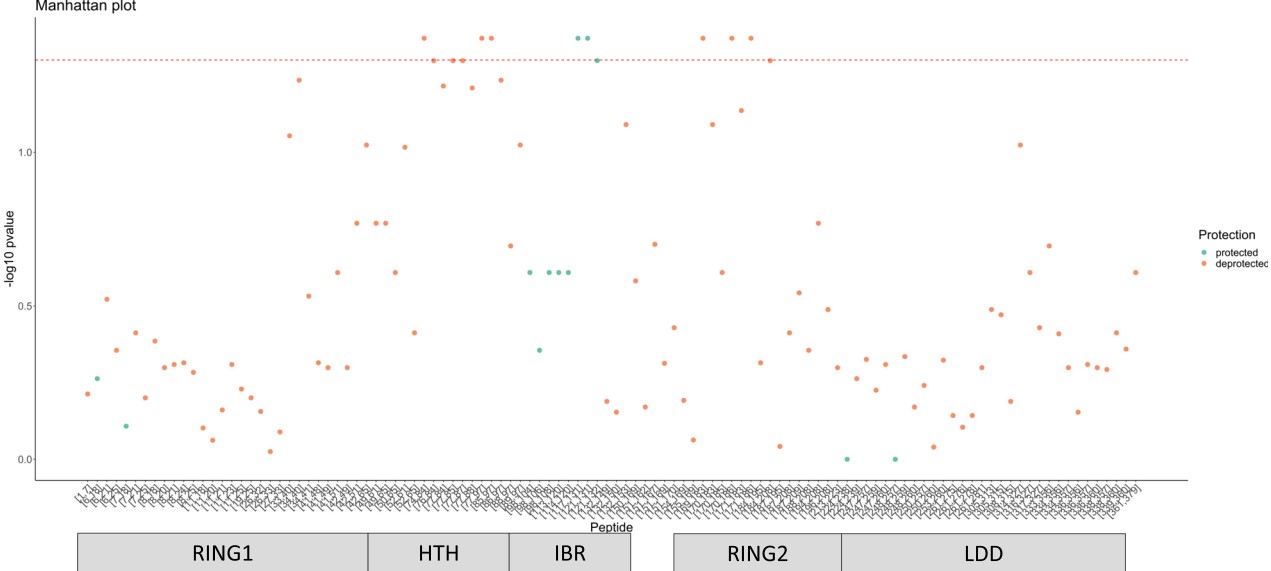

**Fig. 6 Manhattan plots provide spatial context of deuterium chances for HOIP-RBR-dAb25.** A Manhattan plot with peptides described as the amino acid (AA) segments of the protein sequence plotted against the $-\log_{10}(p\text{-value})$. Protein domain annotations are provided in the bar below[26]. The red dashed line indicates a significance threshold $-\log_{10}(0.05)$ and $p$-values have been corrected for multiple testing.

works with centroided HDX spectra, we anticipate further improvements by working with raw spectra and identification confidences directly.

## Methods

**Preliminaries**. In hydrogen–deuterium exchange mass spectrometry, we observe isotope distributions for $i = 1,...,n$ peptides at different exposure times $t_1,...,t_m$ to heavy water ($D_2O$). The isotope distributions are a set of $\frac{m}{z}$-Intensity pairs revealing the relative intensities of each peptide isotope. These isotope distributions are frequently summarised into a centroid via an intensity weighted mean of the $\frac{m}{z}$, which we write as $\frac{\bar{m}}{z}$. Since deuterium is heavier than hydrogen, deuterium interoperation leads to positive shifts in $\frac{m}{z}$ and monitoring this change over time and with respect to the state is the standard usage of HDX-MS. In most scenarios data are replicated, so we observe replicates $r = 1,...,R$ and, potentially, a number of conditions denoted $c = 1,...,C$. For example, binding to antibody is an example of a condition. The observations are then

$$y_{icr}(t) = M(t) = \frac{\bar{m}}{z}(t) \times z - z, \tag{1}$$

where $z$ denotes the charge of the precursor ion. Some practitioners normalise with respect to the initial mass $M(0)$, though this is unnecessary and assumes that mass errors are similar over time. This normalisation is an assumption of *homoscedasticity*, which is unlikely to hold in general because as deuterium is incorporated the isotope distribution undergoes isotopic expansion and so $M(t)$ will have different errors for different values of $t$. If we wish to avoid this normalisation, we can include an offset term in our proposed model (see below).

**Methodological summary**. In this section, we present a high-level methodological summary. In differential HDX-MS, the key quantity of interest is the difference in deuteration patterns between conditions. Conditions can include protein interactions, small molecule binding or environmental perturbation; such as changes in temperature. Statistical methods are typically used to detect significant changes in deuteration between conditions. For example, the *t* test incorporates the difference in mean between conditions with the variance in each condition into a test statistic. The statistic is then compared to a t-distribution, to compute a *p*-value. Once all the *p*-values are computed they are corrected for simultaneous testing of many hypotheses (multiplicity), typically using the Benjamini-Hochberg procedure.

However, *t* tests are applied to differential HDX-MS pointwise, meaning a test for each time point. This excessive amount of testing results in an unnecessary number of false-negatives (low statistical power). Our proposed approach performs curve fitting to the kinetics measured by HDX-MS. Since only one test is performed per peptide, this reduces the number of tests performed. By examining how well these curves fit to the data, we can see if there are significant changes in HDX-MS kinetics. In this case, this is quantified by computing the total difference between the observed data and fitted curves, which is a measure of variance. The appropriate test statistic in this case is an *F*-statistic resulting in an *F* test, in which *p*-values are obtained from an *F*-distribution. Furthermore, these variances are computed for every peptide and so we can improve our method by learning a distribution of variances. By incorporating prior information about this distribution, we can improve the power of our method. The following sections proceed with a detailed mathematical description of this process. Furthermore, we provide a tutorial for analysing differential HDX-MS data in the vignettes of the "hdxstats" package.

**Proposed method**. In this section, we describe the functional model we use to model hydrogen–deuterium exchange. The kinetics of HDX follow a well-appreciated logistic-type model. Typically, peptides rapidly incorporate deuterium and plateau at maximum incorporation. Mathematically, the proposed model takes the following form

$$\mu(t) = a(1 - \exp(-bt)) + d. \tag{2}$$

Each parameter is interpreted as follows. Firstly, $d$ denotes the mass at time 0, but note this is inferred so the uncertainty in the value is captured unlike when normalising by $M(0)$. The parameter d models the initial undeuterated mass and can be forced to 0 when data are normalised in our "hdxstats" package. $b$, the rate constant for HDX, controls the time-dependent kinetics, such that larger values of $b$ denote a more rapid increase in mass (and hence deuterium incorporation). Whilst $a$ controls the plateau of the model representing maximum incorporation. It is also useful to consider a slight modification to the above model, in which a Weibull-type model is used:

$$\mu(t) = a(1 - \exp(-bt^q)) + d. \tag{3}$$

The additional parameter $q$ models additional effects with respect to the temporal kinetics. Sometimes, values of $q$ are greater than 1, suggesting more rapid than exponential incorporations at smaller times, and plateauing more quickly than exponential on longer time scales. Mathematically, $q$ allows some flexibility in the inflexion of the kinetics[33] refer to this model as a stretched exponential. Furthermore, whilst back-exchange correction factors were not available for the

datasets considered in this manuscript, methodology and functions to normalise data based on these data are available in our "hdxstats" package.

**Two-sample test**. The proposed model can be turned into a formal test, using tools from functional data analysis. We first suppose that there is no difference in HDX kinetics for a peptide between two conditions. In such a case, a single function would describe the HDX kinetics regardless of the condition. This constitutes the null hypothesis, whilst the alternative is that there is a difference in HDX kinetics. In this case, independent condition-specific models would better describe the data. To formalise this, for each peptide $i$, we would fit the following model

$$\mu_{ic}(t) = a_{ic}(1 - \exp(-b_{ic}t^{q_{ic}})) + d_{ic}, \tag{4}$$

and correspondingly compute the residual sum of squares under the null (subscript 0) and alternative (subscript 1) hypothesis:

$$\text{RSS}_{0,i} = \sum_{c,r,t}(y_{icrt} - \mu_{i0t})^2 \tag{5}$$

$$\text{RSS}_{1,i} = \sum_{c,r,t}(y_{icrt} - \mu_{ict})^2. \tag{6}$$

These equations describe the squared deviation of the observed data from the fitted mean functions. These quantities will be small if the model is a good fit. The relative plausibility of the two models can be described using the *F*-statistic:

$$F_i = \frac{d_{i,2}}{d_{i,1}} \frac{\text{RSS}_{0,i} - \text{RSS}_{1,i}}{\text{RSS}_{1,i}}, \tag{7}$$

which intuitively weighs up the relative fits of the null and alternative models. The values $d_{i,1}$ and $d_{i,2}$ represent the degrees of freedom of the corresponding *F*-distribution. $d_{i,1}$ is given by $p_2 - p_1$, that is the difference between the number of parameters in the alternative model and the null model, for our approach $p_1 = 3$ and $p_2 = 6$ or $p_1 = 4$ and $p_2 = 8$ if the Weibull model is used. Whilst, $d_{i,2} = n_i - p_2$, where $n_i$ is the number of observations of peptide $i$ across the conditions. Finally, *p*-values are obtained from the corresponding *F* test:

$$F_i \sim_{H_0} F(d_{i,1}, d_{i,2}). \tag{8}$$

Given that we are performing $n$ tests, multiple testing corrections should be performed, typically the Benjamini–Höchberg procedure is recommended[29].

**Effect sizes**. In linear models, effect sizes are given by the values of the coefficients of the appropriate covariates. For functional models, there are the number of possible effect sizes. The appropriate effect size is best chosen based on the question of interest. We describe some of the possible effects that can be extracted from the model:

1. Differences in initial (undeuterated) mass: This quantity is described by extracting $d$ and its confidence intervals from the models.
2. Difference in maximum uptake: This quantity is described by extracting $a + d$ and its confidence intervals from the models. The parameter $a$ is often referred to as the deuterium recovery.
3. Difference in rate kinetics: This quantity is described by extracting $b$ as well as $p$ and their confidence intervals from the models.
4. The global difference between the two models. This quantity is given by the maximum difference between the condition-specific curves:

$$\Delta_{i,\max} = \sup_t |\mu_{i1}(t) - \mu_{i2}(t)| \tag{9}$$

or the integral as suggested by[34]

$$\Delta_{i,\text{int}} = \int_t |\mu_{i1}(t) - \mu_{i2}(t)| dt. \tag{10}$$

5. Local differences between models; that is, an effect at specific time $t_*$. This quantity is given by the difference between the condition-specific curves at that time:

$$\Delta_{i,t_*} = \mu_{i1}(t_*) - \mu_{i2}(t_*). \tag{11}$$

**Empirical Bayes**. Estimates derived from a small number of replicates and timepoints can lead to unstable inferences, as is typically observed in microarray[22] and RNA-seq experiments[23]. To improve stability, and hence power, we propose an empirical Bayes extension to our model. The idea is to shrink estimates of the sample variance towards a pooled estimate. This constituent is a bias-variance trade-off, where we trade a small amount of bias for increased precision. Following[35], inference can then be formulated using a so-called moderated *F*-statistic. To elaborate, let $s_i^2 = \frac{\text{RSS}_{i,1}}{d_{i,1}}$, we can then use this set of variances to identify a global $s_0^2$ and shrink our estimate $s_i^2$ towards $s_0^2$. We assume true variances $\sigma_i^2$ are drawn from the

following scaled inverse $\chi^2$ distribution:

$$\frac{1}{\sigma_i^2} \sim \frac{1}{d_0 s_0^2} \chi^2. \tag{12}$$

It can be shown that[35], the expected value of the posterior of $\tilde{s}_i^2$ is

$$\tilde{s}_i^2 = \frac{d_0 s_0^2 + d_2 s_i^2}{d_0 + d_2}, \tag{13}$$

where the hyperparameters $d_0$ and $s_0^2$ are computed by fitting the following scaled $F$-distribution $s_i^2 \sim s_0^2 F(d_2, d_0)$. Hence, we compute the moderated $F$-statistic with

$$\tilde{F} = \frac{\text{RSS}_{0,i} - \text{RSS}_{1,i}}{\tilde{s}_i^2 d_{i,1}}. \tag{14}$$

**Functional ANOVA**. Our model takes the form of a functional analysis of variance (ANOVA). As a result, it allows for covariate-based experimental designs with multiples levels. One such example would be an analysis based on a set of antibodies or different concentrations of small molecules.

**Linear mixed-effects models**. In this section, we summarise the previous methodology for significance testing for hydrogen–deuterium exchange mass spectrometry that explicitly includes temporal components. The approach, using mixed effects models, is an extension of a previous linear modelling approach[13,16]. In our notation, each set of peptide observations is modelled as

$$y_{icr}(t) = \beta_{it} x_t + \beta_{ic} x_c + \beta_{tc} x_t x_c + u_r w_r + \epsilon, \tag{15}$$

where $\beta_{it}$, $\beta_{ic}$ and $\beta_{tc}$ denote the coefficients of the fixed effects for time, condition and their interaction between time and condition for peptide $i$. The replicates are considered as random effects $u_r$. Random effects attempt to model the additional structured variance corresponding to a particular covariate. We note that in this interpretation that time is considered as a factor rather than a continuous quantity. That is, permuting the ordering of time does not change the model and the explicit correlation induced by the temporal dimension is not modelled. In the case that time is interpreted as a continuous quantity, the model becomes

$$y_{icr}(t) = \beta_{it} t + \beta_{ic} x_c + \beta_{tc} t x_c + u_r w_r + \epsilon, \tag{16}$$

however, HDX data are rarely linear in time and so the data have to be linearised. This can be performed by transforming time according to a shifted log transform: $t \to \log(t + \delta)$. Where $\delta$ is chosen to avoid taking the log of 0. Though this approach reduces the number of tests performed and models the temporal dimension it is not recommended as it can lead to uncontrolled $p$-values and unstable parameter estimates.

**Simulations**. This section describes our proposed simulation study. We begin by sampling, uniformly at random, the number of exchangeable amides of a peptide from between 5 and 25. The sampled number is the number of exchangeable amino acids in the peptide and we sample that number of amino acids from the 20 canonical amino acids with replacement. We then define time points at which to obtain data: $T = \{t_1, \ldots, t_m\}$, with $t_1 = 0$ and $t_i < t_j$ for $i < j$. For time $t_1$, we simulate the undeuterated isotope distribution using a binomial model. For a subsequent time point $t_i$ we sample the percentage incorporation by first sampling from a $m - 1$-variate Dirichlet distribution with concentration parameter $\alpha$, where $\alpha_i = 20/(i - 1)$. From this we obtain a vector $\pi$ which sums to 1. We use the cumulative distribution of $\pi$ as the schedule of incorporations. That is the incorporation at $t_i D_i = \sum_{r=1}^{i-1} \pi_r$ for $i > 1$. This ensure that incorporation is non-decreasing in time. To simulate the effect of a condition, for each time point, we sample an indicator $z_{t_i} \in \{0, 1\}$ such that the $p(z_{t_i} = 0) = 0.95$. If $z_{t_i} = 1$, then we re-sample the incorporation amount and continue on the simulation process. This ensures that roughly 95% of the scenarios have no effect with respect to the condition. A binomial model is used to generate deuterated spectra, where the exchangeable hydrogens are randomly replaced with deuterium according to the incoperation percentage. The isotope distribution simulations are repeated $R$ times to allow for replicates. Centroids summarising the average peptide mass are then computed from the isotope distribution. The centroids are then further corrupted by Gaussian noise, using $\mathcal{N}(0, 0.05)$. In all cases, we simulate 500 measured peptides. We perform simulation scenarios as follows:

- (Scenario 1) 4 time points, 3 replicates and 2 conditions
- (Scenario 2) 4 time points, 2 replicates and 2 conditions
- (Scenario 3) 5 time points, 2 replicates and 2 conditions
- (Scenario 4) 6 time points, 2 replicates and 2 conditions
- (Scenario 5) 6 time points, 2 replicates, 2 conditions and 5% missing values
- (Scenario 6) 6 time points, 2 replicates, 2 conditions, 5% missing values, $p(z_{t_i} = 0) = 0.99$

All simulations are performed 10 times and the distributions compared.

**Performance metrics**. We use the F-score to assess the performance of the different statistical methods. The F-score is the harmonic mean of the precision and recall. Precision is defined as $\frac{tp}{tp + fp}$ and recall is $\frac{tp}{tp + fn}$, where $tp$ is true positives, $fp$ is false positives and $fn$ is false negatives. In words, the F-score weighs how many of the selected peptides are relevant and how many of the relevant peptide are selected.

**Implementation**. The Weibull-type model is implemented using the Levenberg-Marquardt algorithm, an iterative procedure that interpolates between the Gauss-Newton algorithm and gradient descent[36]. The parameters are all constrained to be non-negative and the algorithm ends after 500 iterations or the difference between the successive sum of square residuals is less than $10^{-8}$.

**Residue-level analysis**. For visualisation purposes, we propose a residue-level analysis in the following manner. For each residue $j = 1, \ldots, J$, where $J$ is the total amino acids in the protein, we have a set of $p$-values $j_p$, for $p = 1, \ldots, P_j$ where $P_j$ is the sequence coverage at that amino acid. A residue level $p$-value for residue $j$, for visualisation purposes, is taken as the harmonic mean of the $p$-values $\left\{ j_1, \ldots, j_{P_j} \right\}$. The primary visualisation is a heatmap of the $-\log_{10}$ of this computed value.

**Reporting summary**. Further information on research design is available in the Nature Research Reporting Summary linked to this article.

## Data availability
Data to reproduce the figures are provided in the supplementary material. Experimental data are available from the original manuscripts. Data to reproduce the figure have been deposited on Zenodo: (https://doi.org/10.5281/zenodo.6408572)

## Code availability
Code is available as part of the R-package "hdxstats": https://github.com/ococrook/hdxstats.

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

## Acknowledgements

The authors thank Nathan Gittens for critical reading of the manuscript. O.M.C. acknowledges funding from GSK and a Todd-Bird Junior Research Fellowship from New College Oxford and EPSRC grant number EP/R511742/1. C.W.-C. is an employee of GSK.

## Author contributions

O.M.C. conceived and developed the methodology. O.M.C. wrote the software. O.M.C. analysed and interpreted the results. O.M.C. wrote the manuscript. C.-W.C. and C.M.D. supervised the project and, also, discussed the results and edited the manuscript. All authors shaped the research agenda.

## Competing interests

The authors declare the following competing interests: C.W.C. is an employee of G.S.K.
