## [Peer Review File · Communications Biology]

Reviewers' comments:

Reviewer #1 (Remarks to the Author):

Crook et al correctly assess the current landscape of statistical applications used to determine the significant differences in HDX-MS data, namely that of t-tests and MEMHDX. They accurately note the deficiencies of both systems, most notably the improper use of t-tests in a system where standard deviations are difficult to estimate. Their functional model essentially generates a deuterium uptake model which incorporates a temporal dimension and estimates variance in the observed data using a pooled estimate which provides a more precise estimate of variance with limited observations for any given peptide. Incorporating this within an R-package ensures this tool will likely be of use to researchers in the field.

Minor Points:

- Although a major step forward, especially in the analysis of data that lack replication, the failure to incorporate the spatial aspect of HDX data into the model certainly undercuts the power of the approach. For example, in figure 6, it is plain to see that there is likely a region of protection around residues 98-120. Falling under the threshold of significance, this is likely represents a false negative. Can the authors comment on whether they have attempted adding in a spatial parameter to their model?
 - The authors focus primarily on the elimination of type 1 errors using both their simulated and observed data. Could the authors comment on the ability of t-tests/LMM and their functional approach on the identification of type 2 errors? I understand that the F-score incorporates false-negative identification in the 'recall' aspect, but could this be made more explicit in the text?
- Typos/Suggests

In all figures, could the legends please be made larger? My eyes are (relatively) young and I still struggled with the typeface.

Figure 6: The dashed line should be labeled as the significance threshold.

2.2.2 page 11 "deuterium kinetic were altered. Six of these peptides"

Reviewer #2 (Remarks to the Author):

The authors report a new method to detect differences between reference and perturbed protein samples investigated by HDX-MS. These kinds of difference analysis methods are the most common way to analyse HDX-MS data with numerous methods currently available. The selling point of the authors' method is that it uniquely characterises the temporal dimension by modelling peptide pairs and then evaluating each peptide's correspondence to the model. They have compared their method to existing tools by applying it to experimental data and it is apparent that there's been improved accuracy.

One important thing the authors have not mentioned in their article is the effect on their model of extraneous exchange. The fitting algorithm does not recognise that deuterium incorporation can be erroneously high or low for each peptide and is also unique for each peptide. I appreciate that looking at pairs of peptides should effectively control for this but problems emerge when information is "borrowed across peptides" because 2 neighbouring peptides can have dramatically different experimental isotope uptake because of different structures and column interactions during chromatography. How are these scenarios handled by the authors' method?

On a similar vein the problem with fitting pairs of peptides individually is that there is no appreciation of self-consistency which can only be obtained from a global model. If peptide "A" shows more rapid deuterium incorporation in the perturbed sample from the model there should be carry over to neighbouring peptides that share many of the same amino acids. However, this might not be the case because the peptides are fitted in pairs independent to the remaining data. Larger problems also emerge because of extraneous exchange. How do the authors' reconcile self-consistency without a global model and with data that has not been corrected for extraneous exchange – which is the case for almost all contemporary HDX-MS practitioners.

Reviewer #3 (Remarks to the Author):

please see attached PDF

Summary of the manuscript

HDX-MS is now widely used to study conformational and dynamic behaviors of proteins. Most commonly, differences in HDX rate between different protein states are used to provide a "map" of locations where structure/dynamics are different between states. As such, the identification of significant differences is of considerable interest in the field and the past few years have resulted in considerable activity in the HDX field around the statistical analysis of the HDX-MS data. Much of this work has focused on the analysis of differences at individual HDX labeling times. The present work instead uses a functional data analysis approach to parameterize the HDX uptake curves in order to test significance at the peptide level rather than at the level of individual HDX times. A problem that arises is false positives due to the simultaneous testing of many hypotheses, so the authors propose an empirical Bayesian approach to control false positives. The approach was validated using simulations and existing HDX-MS data sets published by other authors. The authors demonstrate that their approach is superior to other proposed methods.

Overall impression of the work

I must confess that I am not an applied statistician and am not able to fully evaluate the manuscript as it relates to three technical components (1) empirical Bayes estimate of the variance, (2) functional ANOVA, and (3) linear mixed-effects model. Thus, I hope that other reviewers can address those specific areas. Overall, I find the analysis of the simulated and experimental data to be convincing: that the method proposed here is better at controlling false positives while retaining power. I have some reservations about the use and the interpretation of the Weibull model as it relates to the physical processes of HDX and the authors have neglected to cite some key literature. These issues should be addressed in a revised manuscript, but I think with adequate revision this manuscript should be publishable.

Detailed comments (minor issues presented in the following section)

1. Some key literature omitted. During the course of this review, a new paper was published, which unfortunately, covers much of the same ground as the present manuscript. The authors will need to address how their work differs from that of Claesen: "Moderated Test Statistics to Detect Differential Deuteration in Hydrogen/Deuterium Exchange Mass Spectrometry Experiments", J. Claesen, S. Krishnamurthy, A. M. Lau and A. Economou, *Analytical Chemistry* 2021, 10.1021/acs.analchem.1c02346.

Chetty et al, *PNAS* 2009 used the Weibull model to describe HDX kinetics, referred to in that work as a "stretched exponential", well-known in the field of chemical kinetics.

Mazur and Weber, *JASMS* 2017 used functional data analysis methods to test the significance of HDX differences based on an area between the curves approach.

2. Make the paper accessible to the HDX-MS community. The paper reads like it was written by statisticians for other statisticians. This is perfectly acceptable, but if the authors want their methods to gain acceptance within the HDX-MS community (and I think they should!) the authors should consider recasting the text in a way that it can be more comprehensible to mass spectrometrists and structural biologists. It might require writing in a more tutorial style or perhaps adding a section for non-experts.

3. The Weibull model is a phenomenological, not a physical model of HDX kinetics. While many HX curves might appear to follow a logistic rise to maximum (Chetty, et al, *PNAS* 2009), the underlying physical model is a sum of exponentials of the form of equation 2 since each amide unit exchanges by its own set of parameters a and b :

$$d(t) = \sum_{i=1}^n \mu_i (1 - \exp(-a_i t))^{b_i}$$

where n indexes the number of exchangeable amides in the peptide unit and, at least in my formulation, the "d" parameter represents the mass of the undeuterated peptide. Such a model of HDX is hopelessly underdetermined. Thus, the Weibull model is an imperfect

attempt to capture this heterogeneity by using the "p" parameter, a so-called "exponential stretching factor". In most of the work shown in this paper, the number of parameters in the model is nearly equal to the number of distinct physical measurements (HDX times). In addition to not truly being a physical model of HX kinetics, even the Weibull model is potentially under-determined.

The "a" parameter represents the so-called deuterium recovery. In a truly physical model of HDX, the "a" parameter would also be condition-independent since the loss of deuterium is an experimental factor that would be held constant by measurement factors unrelated to the protein condition. Figure 3 illustrates the limitations of the Weibull model. These HDX curves will converge to a common plateau in the limit of long HDX times because the peptide contains the same number of exchanging amides in both conditions. While such HDX times are impractically long, this means that the "a" parameter in the model does not have a physical interpretation. In a physical model, the "a" parameter is best estimated as "deuterium recovery" using a so-called deuteration control (see Masson et al Nature Methods, 2019).

Similarly, the "d" parameter, in a physical model represents the mass of the undeuterated peptide. This will without question be condition-independent. In fact, based on the well-accepted Linderstrom-Lang mechanism of HDX, which is a physical model, the only condition-dependent parameters in the Weibull model would be "b" and "p". Thus, substantial differences in the "a" and "d" parameters indicate that the model is not returning physically meaningful values. I think the authors will discover, as others have reported (see Chetty et al PNAS 2009), that when a and d are constrained to be equal in the two conditions, that two adjustable parameters, "b" and "p" are insufficient to describe the differences in conditions well. Values of the "p" parameter greater than one are also unphysical (again, see Chetty et al, PNAS 2009).

In equations 2 and 3, it is not clear if the authors describe the deuteration of the peptide (mass increment above its undeuterated mass) or its total mass. In the first case, $d = 0$ since it represents the amount of deuteration at $t = 0$, which should be zero by definition. If the latter, then it should exactly equal the mass of the undeuterated peptide. In figure 3, it is reassuringly close to zero, but if unconstrained, the physicality of this model can deteriorate.

Nevertheless, there is potentially some value in the Weibull model as a phenomenological model since it describes several important characteristics of an HDX uptake plot: initial value, plateau, midpoint, and transition steepness, but as the forgoing points make clear it is only a phenomenological model rather than a physical one. However, it may break terribly in cases where the HDX kinetics are clearly biphasic. This is the major point where I see revision is needed. It seems to me that even if not physical, the model can still be used to identify significant differences in HDX behavior between conditions and the paper supports this finding with its results.

4. The simulated noise is too small. The addition of Gaussian noise with a standard deviation of only 0.05 does not represent the variance of real HDX-MS measurements. a standard deviation of 0.1 or even 0.2 is more realistic. The Hageman data was 0.07 and this is pretty small compared to most HDX work out there where 0.20 is a better estimate of the standard deviation. The authors should show with an additional simulation that their method still outperforms others with more realistic data.

Minor issues to correct

p. 2: the solvent accessibility model of HDX is not widely accepted. Linderstrom-Lang theory is preferred (see James et al, Chem Rev 2021).

figure 1: please show all values from simulation or means with standard deviation bars.

figure 1: from which scenario (1,2 3) were figures 1a and 1b drawn?

Figure 1 c: the caption states that the critical region is colored green, but in the figure, the critical region is marked in magenta, not green and the non-critical region is green

figure 1 caption: insert "the reader to" in "We refer [the reader to] the methods for details."

p. 7, figure 3, p 15: the authors have parameterized the physical model using the parameter "p". This symbol is a poor choice in paper that concerns statistical significance. To avoid confusing the reader, I suggest that the authors choose different symbol for the stretching parameter in the Weibull model. Perhaps lambda?

p. 7: Those familiar with chemical kinetics will recognize parameter "b" as the so-called "rate constant". It is traditionally parameterized with k in the HDX literature and more generally in

literature on kinetics.

figure 3: remove "_2" in the titles for plots a and b

figure 3: please define the bars displayed in panels c and d.

figure 3: effect size has not been defined in the manuscript.

figure 4: remove "_2" from plot titles.

figure 6: for completeness, please define in the caption or on the plot the horizontal reference line. is it $p = 0.05$?

Figure 6: I noticed something about this plot: all of the data points above the reference line and those on the reference line are each exactly aligned to single p-values. This might just be related to computational precision in the $\log(p)$ value. Perhaps it is an error of some sort?

p. 15: Many practitioners normalize based on the theoretical mass of the undeuterated peptide. The error in the theoretical mass, based on metrologic error in the atomic weights of the elements, is many orders of magnitude smaller than the errors in HDX-MS measurements and thus can be neglected.

p., 15: replace "constituents" in "This [constitutes] the null hypothesis . . ."?

p. 16: replace "These equation" with "These equations"

p. 18: the simulation actually samples peptides with between 5 and 25 exchangeable amides, rather than lengths between 5 and 25. There are several reasons for this including back-exchange and the impossibility of exchange at proline residues. The authors are referred to the recent review by James et al. Chem Rev 2021. There are no practical consequences here since as far as I can tell, the actual number of amino acid residues in the peptide has no influence on the outcome of the simulation.

p. 18: replace "binomial modal" with "binomial model"?

in the references: Smyth 2004, replace "forassessing" with "for assessing"

in the references: Zhang 2020 citation is incomplete

References section needs carefully proofreading for proper use of upper- and lowercase text.

To the Editor and Reviewers of

“Empirical Bayes functional models for hydrogen deuterium exchange mass spectrometry”

We are grateful for the time and expertise of the editor and the reviewers who have considered our manuscript. The reviewers’ feedback raised a number of valuable points, which can be summarised as follows:

- 1) Global or spatial models of HDX-MS data.
- 2) An examination of false-negatives.
- 3) Extraneous factors.
- 4) Missing literature.
- 5) Accessibility to the HDX-MS audience.
- 6) Interpretation of the model.
- 7) Simulations with increased noise.

We believe that all these points were incredibly valuable to our manuscript and hence we have attempted to fully incorporate the reviewers’ feedback. At a high level, this includes a number of changes. First we edited some sections of the manuscript for clarity. Secondly, we describe to the reviewers several strategies for developing a global or spatial model of HDX-MS data that we attempted. These attempts reveal the complexity of developing a global or spatial model of HDX-MS data, which we feel is beyond the confines of the current manuscript.

In all simulations, we now also report sensitivities and include discussions of false-negatives. We have also included additional discussions of extraneous factors and that back-exchange correction is available as part of our package. We have also added citations to more clearly differentiate ourselves from prior work. We have included a new “methodological summary” section, targeted at the readers without a statistical background. Furthermore, we include tutorials in our package vignettes. We have clarified the interpretation of our model and also demonstrated that our package can fit a number of models, beyond those suggested in the original manuscript. We have added additional simulations to cover scenarios with more noise. We believe that our manuscript is considerably improved and is appropriate to the wide readership of *Communications Biology*.

We attach a point-by-point response to the reviewers suggestions. Our comments specifically to the reviewers are written in blue and the edited text in red. We use square brackets [.] to denote text that was the same as the previous manuscript but was required to identify the location of the edited text. A marked up copy of our manuscript is also attached with new text marked in red. There is an R-package at the following address: <https://github.com/ococrook/hdxstats>.

Yours faithfully,

Dr. Oliver Crook, Todd-Bird Junior Research Fellow in Biochemistry

On behalf of the other authors

Dr. Chun-wa Chung and Prof. Charlotte Deane

Reviewer 1

Crook et al correctly assess the current landscape of statistical applications used to determine the significant differences in HDX-MS data, namely that of t-tests and MEMHDX. They accurately note the deficiencies of both systems, most notably the improper use of t-tests in a system where standard deviations are difficult to estimate. Their functional model essentially generates a deuterium uptake model which incorporates a temporal dimension and estimates variance in the observed data using a pooled estimate which provides a more precise estimate of variance with limited observations for any given peptide. Incorporating this within an R-package ensures this tool will likely be of use to researchers in the field.

We thank the reviewer for their time in evaluating and providing feedback on our manuscript. The reviewer has raised valuable points which we have addressed in full below.

Minor Points:

- Although a major step forward, especially in the analysis of data that lack replication, the failure to incorporate the spatial aspect of HDX data into the model certainly undercuts the power of the approach. For example, in figure 6, it is plain to see that there is likely a region of protection around residues 98-120. Falling under the threshold of significance, this is likely represents a false negative. Can the authors comment on whether they have attempted adding in a spatial parameter to their model?

The author makes an excellent point about not only including the temporal correlation in the model but also the spatial correlations in the amino acid dimension of the data. In the current manuscript we wrote:

“Our approach has a number of limitations. Firstly, we do not model correlations in the spatial domain of HDX data - that is between overlapping peptides. This manifests as clustering of p -values for overlapped peptides.”

Indeed, we agree that not including this correlation likely inflates false-negatives. During the development of our method “hdxstats”, we explored the inclusion of spatial random effects. We considered two strategies, one where a random effect is directly added to the parameters, for example the plateau parameter a , could be modelled as

$$\log(a) = a_{\text{condition}} + u_a,$$

Where $a_{\text{condition}}$ is a fixed effect according to the condition and u_a is a spatial random effect. We can then let $u_a \sim N(0, \Sigma_a)$, where Σ_a is a spatially parameterised covariance matrix, i.e. the entries (i,j) represent the covariance between peptide i and peptide j for plateau parameter. To make the problem tractable, Σ_a should be restricted to overlapping peptides (else the covariance matrix is too large to estimate). That is there is $\Sigma_{a,r}$ where r indexes over groups of overlapping peptides. We then parameterise Σ_a using a peptide similarity function. That is Σ_a is a function of the sequence overlap of the modelled peptides. Though such an approach appeared elegant, we found that fitting these

nonlinear random effects models to not be robust in practice. Indeed, we often found that the estimated Σ was singular and hence the estimated covariances were unreliable. We believe that this strategy is possible but requires advanced statistical methodology well beyond the scope of the current manuscript. The other modelling strategy we investigated was to include a linear random effect added to the nonlinear function, however was found to be an inappropriate model.

Another strategy would be to link the parameters of neighbouring peptides via a hidden-markov model. Whilst a potentially promising approach this would be a serious endeavour even outside the context of hydrogen-deuterium exchange data and thus we have not, yet, pursued this approach.

We also examined alternative approaches which may aid users when power is low. We explored additional methodology based on multi-level testing procedures. Instead of including a spatial effect in the model, we combine p-values after model fitting in the spatial dimension. The results of which are shown in the following figure, as applied to our epitope mapping experiment.

Here the methodology combines p-values in the spatial direction in a windowing strategy. We control the strong-sense family wise error rate and recompute the spatial region (x-axis) that these values cover. The negative logarithm to the base 10 of the p-value is plotted on the y-axis and the dashed line indicated significance at level = 0.05.

While this methodology is hopeful, we found it challenging to interpret and robustly select the correct window size. The interpretation is challenging because this approach combines p-values with protection and deprotection and so physical meaning is lost. The success of this approach also depends on the peptide redundancy of the data. Whilst, we believe there is merit in this approach, without significant development it could be used to once again full development is beyond the scope of this manuscript.

We share the reviewer's desire for a fully spatial HDX model, but we hope that they agree that this is beyond the scope of the present manuscript. We have added the following text to the manuscript:

Several strategies exist to reconcile the spatial dimension of HDX data, including combining p-values using multi-level testing or spatial random effects, which we will consider in future work.

- The authors focus primarily on the elimination of type 1 errors using both their simulated and observed data. Could the authors comment on the ability of t-tests/LMM and their functional approach on the identification of type 2 errors? I understand that the F-score incorporates false-negative identification in the ‘recall’ aspect, but could this be made more explicit in the text?

We agree with the reviewer that we could have been more explicit in the manuscript about false-negatives. We felt that false-negatives outside the context of false-positives are somewhat misleading. That is, calling everything interesting will minimise false-negatives and calling nothing interesting will reduce the false positive rate. Indeed, this is the exact problem with the linear mixed model in the former case and the t-test in the latter case. Our approach correctly balances these quantities, achieving the appropriate middle ground, as demonstrated by the F1 score. To be more explicit about false-negatives, we have computed the sensitivity for each simulation and each method (see plot below). In general, we can see that the LMM approach has the best sensitivity (at the cost of inflating false positives). Improving the sensitivity of our approach could be a key element of future work.

The figure has been added to the supplementary material and the following text added.

To examine false-negatives, we compute the sensitivity (true positive rate) of all the approaches across all the simulation settings. We found that the linear mixed model (LMM) approach has the highest sensitivity across all the approaches. However, this is at the cost of an unacceptable inflation of false-positives (see main text.). The sensitivity of our approach could be improved by switching to a fully Bayesian approach.

Typos/Suggests

In all figures, could the legends please be made larger? My eyes are (relatively) young and I still struggled with the typeface.

We apologise for the small legend text sizes. These have changed so that the smallest type font is now size 14, where possible we have made it significantly larger.

Figure 6: The dashed line should be labeled as the significance threshold.

Thank you for spotting this prior lack of clarity. We have added the following text to the caption:

The red dashed line indicates a significance threshold of $-\log_{10}(0.05)$.

2.2.2 page 11 “deuterium kinetic were altered. Six of these peptides”

Thank you for spotting this typo. We have altered the text to:

deuterium kinetics were altered. Six of these peptides...

Reviewer 2:

The authors report a new method to detect differences between reference and perturbed protein samples investigated by HDX-MS. These kinds of difference analysis methods are the most common way to analyse HDX-MS data with numerous methods currently available. The selling point of the authors' method is that it uniquely characterises the temporal dimension by modelling peptide pairs and then evaluating each peptide's correspondence to the model. They have compared their method to existing tools by applying it to experimental data and it is apparent that there's been improved accuracy.

We thank the reviewer for their positive review of our work and the comment that it advances over currently available tools in the literature.

One important thing the authors have not mentioned in their article is the effect on their model of extraneous exchange. The fitting algorithm does not recognise that deuterium incorporation can be erroneously high or low for each peptide and is also unique for each peptide. I appreciate that looking at pairs of peptides should effectively control for this but problems emerge when information is "borrowed

across peptides" because 2 neighbouring peptides can have dramatically different experimental isotope uptake because of different structures and column interactions during chromatography. How are these scenarios handled by the authors' method?

The reviewer raises a valuable point about extraneous exchange which we address below.

We agree with the reviewer that looking at pairs of peptides reduces the problem of extraneous exchanges, because these factors should affect the peptides equally. Otherwise, there is something wrong with the experimental design that no amount of statistical methodology can overcome. We appreciate that there may be problems with borrowing information across peptides and indeed modelling the uptake of groups of peptides can be challenging. In our approach, the borrowing of information happens after the model fitting at the level of the residuals (the error remaining after modelling). We then obtain residuals for each peptide and our borrowing of information estimates a distribution for these residuals. A simple application of Bayes' theorem allows us to update the relevant statistics, but note these are not the parameters of the individual uptake plots. Since we estimate a distribution, these residuals will take on a range of values often due to these extraneous factors that the reviewer mentions. By borrowing information in this manner, we are accounting for the residual variance that is unexplained by the original model fits (perhaps because of extraneous factors). We agree with the reviewer that there would be problems if we were borrowing information at the level of parameters of the model. Since we are not, such problems are minimal.

If a fully deuterated sample is provided users can normalise their data using our package, using the `normalisehdlx` function, which allows back-exchange correction and exchangeable amide correction. The reviewer is welcome to open a comment on our github page if they wish to see a further approach incorporated. We have added the following comment to the methods section:

Furthermore, whilst back-exchange correction factors were not available for the datasets considered in this manuscript, methodology and functions to normalise data based on these data are available in our hdxstats package.

On a similar vein the problem with fitting pairs of peptides individually is that there is no appreciation of self-consistency which can only be obtained from a global model. If peptide "A" shows more rapid deuterium incorporation in the perturbed sample from the model there should be carry over to neighbouring peptides that share many of the same amino acids. However, this might not be the case because the peptides are fitted in pairs independent to the remaining data. Larger problems also emerge because extraneous exchange. How to the authors' reconcile self-consistency without a global model and with data that has not been corrected for extraneous exchange – which is the case for almost all contemporary HDX-MS practitioners.

This is an excellent point raised by the reviewer and we share their concerns. A global model which we refer to as a model with spatial parameters would indeed improve sensitivity and aid in interpretation of hdx data. Initially, in our development of our approach we sought to incorporate a spatial random effect, which would effectively account for differences in neighboring peptides. However, we found this

approach lacked robustness due to poor estimation of the relevant covariance matrices. Many practitioners also resort to qualitative approaches to improve robustness, for example using at least 2 overlapping peptides to declare significance. Though this heuristic is likely useful, it will generate false negatives.

Reviewer 3:

Summary of the manuscript

HDX-MS is now widely used to study conformational and dynamic behaviors of proteins. Most commonly, differences in HDX rate between different protein states are used to provide a “map” of locations where structure/dynamics are different between states. As such, the identification of significant differences is of considerable interest in the field and the past few years have resulted in considerable activity in the HDX field around the statistical analysis of the HDX-MS data. Much of this work has focused on the analysis of differences at individual HDX labeling times. The present work instead uses a functional data analysis approach to parameterize the HDX uptake curves in order to test significance at the peptide level rather than at the level of individual HDX times. A problem that arises is false positives due to the simultaneous testing of many hypotheses, so the authors propose an empirical Bayesian approach to control false positives. The approach was validated using simulations and existing HDX-MS datasets published by other authors. The authors demonstrate that their approach is superior to other proposed methods.

We thank the reviewer for the detailed summary and evaluation of our manuscript.

Overall impression of the work

I must confess that I am not an applied statistician and am not able to fully evaluate the manuscript as it relates to three technical components (1) empirical Bayes estimate of the variance, (2) functional ANOVA, and (3) linear mixed-effects model. Thus, I hope that other reviewers can address those specific areas. Overall, I find the analysis of the simulated and experimental data to be convincing: that the method proposed here is better at controlling false positives while retaining power. I have some reservations about the use and the interpretation of the Weibull model as it relates to the physical processes of HDX and the authors have neglected to cite some key literature. These issues should be addressed in a revised manuscript, but I think with adequate revision this manuscript should be publishable.

We thank the reviewer for their time and expertise when reviewing our manuscript and for their valuable feedback. The reviewer has raised a number of instructive comments and we address these in full in our comments below.

Detailed comments (minor issues presented in the following section)

1. Some key literature omitted. During the course of this review, a new paper was published, which unfortunately, covers much of the same ground as the present manuscript. The authors will need to

address how their work differs from that of Claesen: “Moderated Test Statistics to Detect Differential Deuteration in Hydrogen/Deuterium Exchange Mass Spectrometry Experiments”, J. Claesen, S. Krishnamurthy, A. M. Lau and A. Economou, *Analytical Chemistry* 2021, 10.1021/acs.analchem.1c02346. Chetty et al, PNAS 2009 used the Weibull model to describe HDX kinetics, referred to in that work as a “stretched exponential”, well-known in the field of chemical kinetics. Mazur and Weber, JASMS 2017 used functional data analysis methods to test the significance of HDX differences based on an area between the curves approach.

We thank the reviewer for highlighting some additional literature. The paper of Claesen indeed covers some of the same ground but their approach is well-known in transcriptomics and proteomics literature, where linear models are more appropriate. The main novelty of our method is the appropriate application of statistics for a non-linear model. The empirical Bayes is a small part of the picture for our method and this combination with a non-linear model is novel for HDX. We also see the development of an R-package to analyze the data as fundamental. The method of Claesen et al is purely an advertisement for statistical methods that already exist i.e. via the *limma*, *DEP*, *MSqROB*, *DqMS* etc R-packages. Furthermore, they still use t-tests which, as we highlighted in the manuscript, are not applicable to the applications we are interested in. We have added the following text to clarify:

We establish this method for HDX data as applied to functional models, see Claesen et al (2021) for applications to linear HDX models.

The paper of Chetty et al. is certainly important and we regret omitting the citation. We have added the citation to the methods section:

Chetty et al. refer to this model as a stretched exponential.

The paper of Mazur and Weber JASMS 2017 is certainly interesting and has a similar vein to our approach. However, we could not find a usable implementation of their approach and reimplementing their method is beyond the scope of this article. Their approach is related to our method but instead of using the statistics derived from the deviance from the fitted curve they compute the area between curves. In our paper, we encourage the use of the area between curves as a useful effect size as we already suggested in the article. We briefly describe why their method is not the appropriate statistic in this setting. Firstly, the method relies on computing the standard error of the area and then plugin these into a t-statistic. The fundamental issue with this approach is that additional variance is introduced unnecessarily through the accuracy of the numerical integration. Typically, such additional variance introduces false positives and false negatives. Furthermore, it is not clear these statistics will be t-distributed. In their supplementary they make a number of incorrect arguments supporting a t-statistic. The first of which is that they check that the standard errors are normally distributed, in fact they want to check that the compute areas are normally distributed. The second is that normality is checked via a Shapiro-Wilke test, but this is a fundamental error. This test can only reject normality, not provide evidence for it. Our approach avoids these unnecessary and incorrect steps.

In our revised manuscript, we write in the effect size section:

as suggested by Mazur and Weber 2017.

2. Make the paper accessible to the HDX-MS community. The paper reads like it was written by statisticians for other statisticians. This is perfectly acceptable, but if the authors want their methods to gain acceptance within the HDX-MS community (and I think they should!) the authors should consider recasting the text in a way that it can be more comprehensible to mass spectrometrists and structural biologists. It might require writing in a more tutorial style or perhaps adding a section for non-experts.

This is an important piece of feedback from the reviewer and we certainly wish to make our paper accessible to the HDX-MS community. In particular, we wished to remain precise without compromising communications. However, we obviously previously did not quite hit the mark here. We believe a tutorial style is best left to our package `hdxstats` vignettes, which walk users through the data analysis in a step-by-step fashion, which includes customizable graphics:

<https://github.com/ococrook/hdxstats/tree/main/vignettes>. The package is organic and so we hope to keep adding analysis of challenging datasets to this package and expanding the tutorials. Furthermore, future methodological updates will be added to this package maintaining a coherent style.

In addition, we now include a “methodological summary” section, which provides a description of our method whilst only assuming basic familiarity with statistical ideas. The added text is following and we hope the combination of vignette and additional text makes the paper more accessible:

In this section, we present a high-level methodological summary. In differential HDX-MS, the key quantity of interest is the difference in deuteration patterns between conditions. Conditions can include protein interactions, small molecule binding or environmental perturbation; such as, changes in temperature. Statistical methods are typically used to detect significant changes in deuteration between conditions. For example, the t-test incorporates the difference in mean between conditions with the variance in each condition into a test statistic. The statistic is then compared to a t-distribution, to compute a p -value. Once all the p -values are computed they are corrected for simultaneous testing of many hypotheses (multiplicity), typically using the Benjamini-Hochberg procedure.

However, t-tests are applied to differential HDX-MS pointwise, meaning a test for each time point. This excessive amount of testing results in an unnecessary number of false-negatives (low statistical power). Our proposed approach performs curve fitting to the kinetics measured by HDX-MS. Since only one test is performed per peptide, this reduces the number of tests performed. By examining how well these curves fit to the data, we can see if there are significant changes in HDX-MS kinetics. In this case, this is quantified by computing the total difference between the observed data and fitted curves, which is a measure of variance. The appropriate test statistic in this case is an F -statistics resulting in an F -test, in which p -values are obtained from an F -distribution. Furthermore, these variances are computed for every peptide and so we can improve our method by learning a distribution of variances. By incorporating prior information about this distribution, we can improve the power of our method. The following sections proceed with a detailed mathematical description of this process. Furthermore, we provide a tutorial for analysing differential HDX-MS data in the vignettes of the “`hdxstats`” package.

3. The Weibull model is a phenomenological, not a physical model of HDX kinetics. While many HDX curves might appear to follow a logistic rise to maximum (Chetty, et al, PNAS 2009), the underlying physical model is a sum of exponentials of the form of equation 2 since each amide unit exchanges by its own set of parameters a and b :
$$f(t) = \sum_{i=1}^n \frac{1}{n} \left(1 - \exp(-a_i t) \right)^{b_i} \frac{d_i}{d} + \frac{a_i}{a} - \frac{b_i}{b} \sum$$
 where n indexes the number of exchangeable amides in the peptide unit and, at least in my formulation, the “ d ” parameter represents the mass of the undeuterated peptide. Such a model of HDX is hopelessly underdetermined. Thus, the Weibull model is an imperfect attempt to capture this heterogeneity by using the “ p ” parameter, a so-called “exponential stretching factor”. In most of the work shown in this paper, the number of parameters in the model is nearly equal to the number of distinct physical measurements (HDX times). In addition to not truly being a physical model of HDX kinetics, even the Weibull model is potentially under-determined.

We agree with the reviewer that we did not communicate the purpose of “physical” here. Our language was intended to avoid conflation with a non-parametric model, since most functional data analysis is non-parametric using splines or fourier transforms. We have edited instances of “physical model” to simply “model” or “functional model” or “parametric functional model”, whenever the context is appropriate. We have also referenced the Chetty paper.

We also agree with the reviewer that these models are potentially underdetermined. However, this is precisely the motivation for our article. We can use statistics to make the trade-off between parameters, model complexities and explained variance. We disagree with the adjective “hopeless”, empirical Bayesian and Bayesian methods can handle scenarios which are underdetermined by incorporating prior information or distributional information. This is exactly the purpose of such statistical methodology. Modelling scenarios where the number of variables exceeds the number of observations is central to most of modern statistics and if embraced by the HDX community should generate further gains.

The “ a ” parameter represents the so-called deuterium recovery. In a truly physical model of HDX, the “ a ” parameter would also be condition-independent since the loss of deuterium is an experimental factor that would be held constant by measurement factors unrelated to the protein condition. Figure 3 illustrates the limitations of the Weibull model. These HDX curves will converge to a common plateau in the limit of long HDX times because the peptide contains the same number of exchanging amides in both conditions. While such HDX times are impractically long, this means that the “ a ” parameter in the model does not have a physical interpretation. In a physical model, the “ a ” parameter is best estimated as “deuterium recovery” using a so-called deuteration control (see Masson et al Nature Methods, 2019). Similarly, the “ d ” parameter, in a physical model represents the mass of the undeuterated peptide. This will without question be condition-independent. In fact, based on the well accepted Linderstrom-Lang mechanism of HDX, which is a physical model, the only condition-dependent parameters in the Weibull model would be “ b ” and “ p ”. Thus, substantial differences in the “ a ” and “ d ” parameters indicate that the model is not returning physically meaningful values. I think the authors will discover, as others have reported (see Chetty et al PNAS 2009), that when a and d are constrained to be equal in the two conditions, that two adjustable parameters, “ b ” and “ p ” are insufficient to describe the differences in conditions well. Values of the “ p ” parameter greater than one are also unphysical (again, see Chetty et al, PNAS 2009).

We agree with the reviewer on these points and have removed references to “physical” in our manuscript. Furthermore, we have clarified the interpretation of a :

The parameter a is often referred to as the deuterium recovery.

As well as the parameter d :

The parameter d models the initial undeuterated mass and can be forced to 0 when data are normalised in our “hdxstats” package.

In equations 2 and 3, it is not clear if the authors describe the deuteration of the peptide (mass increment above its undeuterated mass) or its total mass. In the first case, $d = 0$ since it represents the amount of deuteration at $t = 0$, which should be zero by definition. If the latter, then it should exactly equal the mass of the undeuterated peptide. In figure 3, it is reassuringly close to zero, but if unconstrained, the physicality of this model can deteriorate. Nevertheless, there is potentially some value in the Weibull model as a phenomenological model since it describes several important characteristics of an HDX uptake plot: initial value, plateau, midpoint, and transition steepness, but as the foregoing points make clear it is only a phenomenological model rather than a physical one. However, it may break terribly in cases where the HDX kinetics are clearly biphasic. This is the major point where I see revision is needed. It seems to me that even if not physical, the model can still be used to identify significant differences in HDX behavior between conditions and the paper supports this finding with its results.

The reviewer makes excellent points. The models are sufficiently flexible to handle mass increment, where d is fixed at 0 or total mass where d is inferred. We have found that allowing d to be inferred can be useful if there are undesired mass fluctuations e.g. due to temperature. Allowing d to be inferred also allows errors to be different at time 0 (where they should just be mass-spectrometric) and other times (where there maybe errors from the HDX process).

Indeed, the Weibull model fails in some cases but since it is embedded in a statistical procedure it can't be misleading. That is model misspecification will result in inflated false negatives rather than false positives.

Our accompanying “hdxstats” package was designed to be sufficiently flexible to fit all flavours of functional models and perform the correct statistics. Instead of a Weibull model one could use the sum of logistics $a*(1 - \exp(-b*t)) + c*(1 - \exp(-d*t))$, for example. The model fit for the Weibull (left) and this model (right) are shown below. High resolution images are placed in the supplementary material.

It is almost trivial to fit this model in our package, using the following R code. Indeed, in principle, any functional model could be fitted using the formula notation:

```

res2 <- differentialUptakeKinetics(object = MBPqDF[,1:100], #provide a QFeature object,

```

```
formula = value ~ a*(1 - exp(-b * timepoint)) + c * (1 - exp(-d * timepoint)),  
feature = rownames(MBPqDF)[[1]][37], # which peptide to do we fit  
start = list(a = 0, b = 0.0001, c = 0, d = 0.0001) # what are the starting parameter
```

```
guesses  
````
```

To avoid overfitting and compute statistically whether one model is preferred over another our package also allows the computations of quantities such as log-likelihoods and deviances for model comparison. It is as simple as follows:

```
````{r,}  
logLik(res2)  
````
```

These approaches have been added to our vignette, where we walk users through how to use our package.

We have added the following details to the supplementary materials (section 8.4) for additional clarification:

The accompanying “hdxstats” package also supports further functional modelling strategies. For example, the logistic and Weibull models can be replaced with more exotic models which may be more appropriate to the application. Biphasic exchange may support a sums of logistic model, such as:

$$\mu(t) = a(1 - \exp(-bt)) + c(1 - \exp(-dt)).$$

The statistical methodologies detailed in the main article still apply by simply replacing the functional model for the one above. Furthermore, the package also supports comparison of different models using statistical methods such as the log-likelihood. For demonstration, we fit the sums of logistic model to an example peptide from the MBP dataset (see figure 21) and we show a comparison of this to the fit from the Weibull model (see figure 22).

4. The simulated noise is too small. The addition of Gaussian noise with a standard deviation of only 0.05 does not represent the variance of real HDX-MS measurements. A standard deviation of 0.1 or even 0.2 is more realistic. The Hageman data was 0.07 and this is pretty small compared to most HDX work out there where 0.20 is a better estimate of the standard deviation. The authors should show with an additional simulation that their method still outperforms others with more realistic data.

This is a good point raised by the reviewer. To cover more scenarios, we now vary the standard deviations at 0.1, 0.2 and 0.5 to demonstrate that our model can handle very noisy data. We also varied the number of replicates 2 and 3 and hence have added 6 additional simulations. These have been added to the supplementary material.

The following text has been added to the main text:

We explore simulations with higher levels of noise in the supplementary material and obtain the same conclusions (see supplementary material).

The following text was added to the supplementary.

To test the effect of noise on the various statistical approaches, we varied the standard deviation of the residual noise. We perform the following additional simulation scenarios:

- Standard deviation = 0.1 and 3 replicates (simulation 7)
- Standard deviation = 0.2 and 3 replicates (simulation 8)
- Standard deviation = 0.5 and 3 replicates (simulation 9)
- Standard deviation = 0.1 and 2 replicates (simulation 10)
- Standard deviation = 0.2 and 2 replicates (simulation 11)
- Standard deviation = 0.5 and 2 replicates (simulation 12)

Our simulations demonstrate that even in very noisy data our functional model achieves the highest performance.

Minor issues to correct

p. 2: the solvent accessibility model of HDX is not widely accepted. Linderstrom-Lang theory is preferred (see James et al, Chem Rev 2021).

We have now correctly referenced that the guiding hypothesis for HDX is Linderstrom-Lang theory.

figure 1: please show all values from simulation or means with standard deviation bars.

figure 1: from which scenario (1,2 3) were figures 1a and 1b drawn?

Figure 1 (a,b,c) is an illustrative cartoon of the model, however, this was unclear and we have clarified in the caption. All points are plotted already.

(a,b,c) Cartoon of the proposed approach to statistical analysis of HDX-MS data.

Figure 1 c: the caption states that the critical region is colored green, but in the figure, the critical region is marked in magenta, not green and the non-critical region is green

We apologise that the colours were swapped during compiling of the manuscript and the caption has now been corrected

figure 1 caption: insert "the reader to" in "We refer [the reader to] the methods for details."

We have corrected this caption.

p. 7, figure 3, p 15: the authors have parameterized the physical model using the parameter "p". This symbol is a poor choice in paper that concerns statistical significance. To avoid confusing the reader, I suggest that the authors choose different symbol for the stretching parameter in the Weibull model. Perhaps lambda?

The reviewer is right that p might cause some confusion. We have opted to change all these instances of p to q, whilst lambda might be a good option, we believe it is unnecessary to mix latin and greek alphabets.

p. 7: Those familiar with chemical kinetics will recognize parameter "b" as the so-called "rate constant". It is traditionally parameterized with k in the HDX literature and more generally in literature on kinetics.

We appreciate that the letter "k" is used in the HDX literature but "k" is reserved for the power (stretch parameter) in the statistics literature. To avoid unnecessary conflict, we have chosen to stay with "b" as it is impartial amongst both communities.

figure 3: remove "\_2" in the titles for plots a and b

The underscore indicates the charge state and we apologise for being unclear, since some peptides appear more than once with different charge states we have chosen to keep the notation. We have added a clarifying note to the text.

The number after the underscore indicates the charge state.

figure 3: please define the bars displayed in panels c and d.

We apologise for the lack of clarity here. We have added the following text:

(c,d) The pointer refers to the mean estimate with tails corresponding to the 95% confidence interval.

figure 3: effect size has not been defined in the manuscript.

We apologise for this omission. We have defined and referenced forward to the method section:

the strength of relationship between variables (see methods)

figure 4: remove "\_2" from plot titles.

The underscore indicates the charge state and we apologise for being unclear, since some peptides appear more than once with different charge states we have chosen to keep the notation. We have added a clarifying note to the text.

The number after the underscore indicates the charge state.

figure 6: for completeness, please define in the caption or on the plot the horizontal reference line. is it  $p = 0.05$ ?

Thank you for spotting this lack of clarity. We have added the following text to the caption:

The red dashed line indicates a significance threshold of  $-\log_{10}(0.05)$  and p-values have been corrected for multiple testing.

Figure 6: I noticed something about this plot: all of the data points above the reference line and those on the reference line are each exactly aligned to single p-values. This might just be related to computational precision in the  $\log(p)$  value. Perhaps it is an error of some sort?

This is an interesting observation the reviewer has made but there is no error. Adjusted p-values (FDR) are a set property and so will contain repeats which are typically expected.

To see this more clearly if 1000 p-values are simulated from a uniform distribution. p-values are distributed uniformly between 0 and 1 under the null hypothesis. If we check for duplicates, we should find 0 duplicates amongst these values. However, once we correct for multiple testing the adjusted p-values contain duplicates. The reviewer can check for themselves using the following R code. In fact we note that more than 90% of the p-values were duplicates.

```
> set.seed(1)
> sum(duplicated(runif(1000)))
[1] 0
> sum(duplicated(p.adjust(runif(1000), method = "BH")))
[1] 925
```

p. 15: Many practitioners normalize based on the theoretical mass of the undeuterated peptide. The error in the theoretical mass, based on metrologic error in the atomic weights of the elements, is many orders of magnitude smaller than the errors in HDX-MS measurements and thus can be neglected.

We agree with the reviewer that the error in the undeuterated peptide should be small. However, we have noticed in many datasets that the initial mass error can fluctuate considerably “or drift”. This can often be corrected using lock-mass corrections but in some cases practitioners have not performed such corrections. Our package is flexible enough such that if users are confident they can simply force  $d = 0$ , but we wish to leave the text as is to highlight that the normalisation assumption is also a modelling assumption with which hdx practitioners should be aware. Indeed in sufficiently replicated data, allowing a variable  $d$  to vary even though the error might be small will lead to improved power versus using a normalisation approach.

p. 15: replace "constituents" in "This [constitutes] the null hypothesis . . ."?

Thank you for spotting this typo. It has been corrected.

p. 16: replace "These equation" with "These equations"

Thank you for spotting this typo. It has been corrected.

p. 18: the simulation actually samples peptides with between 5 and 25 exchangeable amides, rather than lengths between 5 and 25. There are several reasons for this including back-exchange and the impossibility of exchange at proline residues. The authors are referred to the recent review by James et al. Chem Rev 2021. There are no practical consequences here since as far as I can tell, the actual number of amino acid residues in the peptide has no influence on the outcome of the simulation.

This is a keen observation by the reviewer and they are correct. We have edited the text such that it now reads

This section describes our proposed simulation study. We begin by sampling, uniformly at random, the number of exchangeable amides of a peptide from between \$5\$ and \$25\$.

p. 18: replace "binomial modal" with "binomial model"?

Thank you for spotting this typo. It has been corrected.

in the references: Smyth 2004, replace "forassessing" with "for assessing"

We have corrected this reference.

in the references: Zhang 2020 citation is incomplete

Thank you, we have updated this reference. When we extracted the citation from google scholar the article did not have a volume number yet.

References section needs carefully proofreading for proper use of upper- and lowercase text.

In the underlying bibliography file the capitalization is correct, the nature style file imposes the erroneous lowercase text which appears in the pdf manuscript. When the journal compiles the underlying .tex files for production the citations will be correct. We do not wish to make any changes as from prior experience this causes confusion when the paper is put into production.

REVIEWERS' COMMENTS:

Reviewer #1 (Remarks to the Author):

I am very much satisfied with the revised manuscript.

Glen Masson

Reviewer #2 (Remarks to the Author):

I am satisfied with the revised article and recommend publication as is.

Reviewer #3 (Remarks to the Author):

The authors have adequately revised the manuscript to address my comments. I find the manuscript suitable for acceptance.

Minor:

p. 15: "The appropriate test statistic in this case is an F-statistics . . ." change last word from plural to singular?

I encourage the authors to add, parenthetically, that the b parameter is known in the HX field as the "rate constant for HX"